# Learning Disentangled Representations for CounterFactual Regression

**Negar Hassanpour & Russell Greiner**
Department of Computing Science
University of Alberta
Edmonton, Alberta, T6G 2E8, CANADA
{hassanpo,rgreiner}@ualberta.ca

## Abstract

We consider the challenge of estimating treatment effects from observational data; and point out that, in general, only some factors based on the observed covariates $X$ contribute to selection of the treatment $T$, and only some to determining the outcomes $Y$. We model this by considering three underlying sources of $\{\,X,\,T,\,Y\,\}$ and show that explicitly modeling these sources offers great insight to guide designing models that better handle selection bias in observational datasets. This paper is an attempt to conceptualize this line of thought and provide a path to explore it further.

In this work, we propose an algorithm to (1) identify disentangled representations of the above-mentioned underlying factors from any given observational dataset $\mathcal{D}$ and (2) leverage this knowledge to reduce, as well as account for, the negative impact of selection bias on estimating the treatment effects from $\mathcal{D}$. Our empirical results show that the proposed method achieves state-of-the-art performance in both individual and population based evaluation measures.

## 1 Introduction

As we rely more and more on artificial intelligence (AI) to automate the decision making processes, accurately estimating the causal effects of taking different actions gains an essential role. A prominent example is precision medicine – *i.e.*, the customization of health-care tailored to each individual patient – which attempts to identify which medical procedure $t \in \mathcal{T}$ will benefit a certain patient $x$ the most, in terms of the treatment outcome $y \in \mathbb{R}$. Learning such models requires answering counterfactual questions (Rubin, 1974; Pearl, 2009) such as: *"Would this patient have lived longer [and by how much], had she received an alternative treatment?"*.

For notation: a dataset $\mathcal{D} = \{\,[x_i,\,t_i,\,y_i]\,\}_{i=1}^{N}$ used for treatment effect estimation has the following format: for the $i^{th}$ instance (*e.g.*, patient), we have some context information $x_i \in \mathcal{X} \subseteq \mathbb{R}^K$ (*e.g.*, age, BMI, blood work, etc.), the administered treatment $t_i$ chosen from a set of treatment options $\mathcal{T}$ (*e.g.*, $\{0: \text{medication}, 1: \text{surgery}\}$), and the respective observed outcome $y_i \in \mathcal{Y}$ (*e.g.*, survival time; $\mathcal{Y} \subseteq \mathbb{R}^{+}$) as a result of receiving treatment $t_i$. Note that $\mathcal{D}$ only contains the outcome of the administered treatment (aka *observed* outcome: $y_i$), but not the outcome(s) of the alternative treatment(s) (aka *counterfactual* outcome(s): $y_i^t$ for $t \in \mathcal{T} \setminus \{t_i\}$), which are inherently unobservable. For the binary-treatment case, we denote the alternative treatment as $\neg t_i = 1 - t_i$.

Pearl (2009) demonstrates that, in general, causal relationships can only be learned by experimentation (on-line exploration), or running a Randomized Controlled Trial (RCT), where the treatment assignment does not depend on the individual $X$ – see Figure 1(a). In many cases, however, this is expensive, unethical, or even infeasible. Here, we are forced to approximate treatment effects from off-line datasets collected through Observational Studies. In such datasets, the administered treatment $T$ depends on some or all attributes of individual $X$ – see Figure 1(b). Here, as $\Pr(\,T\,|\,X\,) \neq \Pr(\,T\,)$, we say these datasets exhibit **selection bias** (Imbens & Rubin, 2015). Figure 2 illustrates selection bias in an example (synthetic) observational dataset.

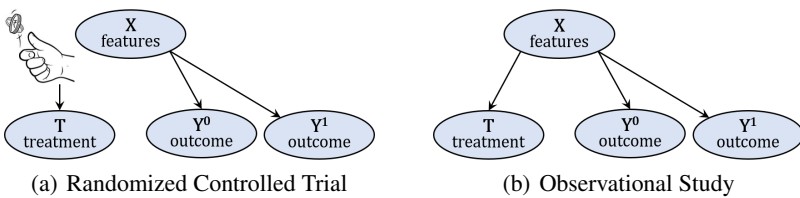

(a) Randomized Controlled Trial      (b) Observational Study

Figure 1: Belief net structure for randomized controlled trials and observational studies. Here, $Y^0$ ($Y^1$) is the outcome of applying $T =$ treatment#0 (#1) to the individual represented by $X$.

Here, we want to accurately estimate the Individual Treatment Effect (ITE) for each instance $i$ – $i.e.$, to estimate $e_i = y_i^1 - y_i^0$. We frame the solution as learning the function $f : \mathcal{X} \times \mathcal{T} \to \mathcal{Y}$ that can accurately predict the outcomes (both observed $\hat{y}_i^{\,t_i}$ as well as counterfactuals $\hat{y}_i^{\,\neg t_i}$) given the context information $x_i$ for each individual. As mentioned earlier, there are two challenges associated with estimating treatment effects:

(i) The fact that counterfactual outcomes are unobservable ($i.e.$, not present in any training data) makes estimating treatment effects more difficult than the generalization problem in the supervised learning paradigm. This is an inherent characteristic of this task.

(ii) Selection bias in observational datasets implies having fewer instances within each treatment arm at specific regions of the domain. This sparsity, in turn, would decrease the accuracy and confidence of predicting counterfactuals at those regions.

This paper addresses the second challenge by investigating the root causes of selection bias, by dissecting and identifying the underlying factors that can generate an observational dataset $\mathcal{D}$, and leveraging this knowledge to reduce, as well as account for, the negative impact of selection bias on estimating the treatment effects from $\mathcal{D}$. In this work, we borrow ideas from the representation learning literature (Bengio et al., 2013) in order to reduce selection bias and from the domain adaptation literature (Shimodaira, 2000) in order to account for the remainder selection bias that (might) still exist after its reduction.

Our analysis relies on the following assumptions:
**Assumption 1: Unconfoundedness** (Rosenbaum & Rubin, 1983) – There are no unobserved confounders ($i.e.$, covariates that contribute to both treatment selection procedure as well as determination of outcomes). Formally, $\{Y^t\}_{t \in \mathcal{T}} \perp\!\!\!\perp T \mid X$.
**Assumption 2: Overlap** (Imbens, 2004) – Every individual $x$ should have a non-zero chance of being assigned to any treatment arm. That is, $\Pr(T = t \mid X = x) \neq 0 \quad \forall t \in \mathcal{T}, \forall x \in \mathcal{X}$.

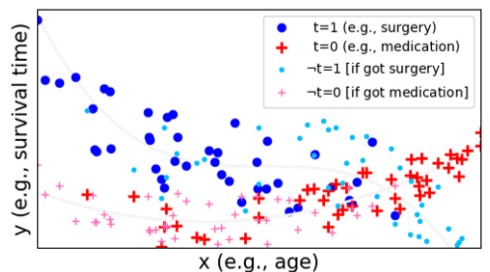

Figure 2: An example observational dataset. Here, to treat heart disease, a doctor typically prescribes surgery ($t = 1$) to younger patients ($\bullet$) and medication ($t = 0$) to older ones ($+$). Note that instances with larger (resp., smaller) $x$ values have a higher chance to be assigned to the $t = 0$ (resp., 1) treatment arm; hence we have selection bias. The counterfactual outcomes (only used for evaluation purpose) are illustrated by small $\bullet$ ($+$) for $\neg t = 1$ (0).

These two assumptions together are called *strong ignorability* (Rosenbaum & Rubin, 1983). Imbens & Wooldridge (2009) showed that strong ignorability is sufficient for ITE to be identifiable.

Without loss of generality, we assume that the random variable $X$ follows a(n unknown) joint probability distribution $\Pr(X \mid \Gamma, \Delta, \Upsilon)$, treatment $T$ follows $\Pr(T \mid \Gamma, \Delta)$, and outcome $Y^T$ follows $\Pr_T(Y^T \mid \Delta, \Upsilon)$, where $\Gamma$, $\Delta$, and $\Upsilon$ represent the three underlying factors[1] that generate an obser-

---

[1] Examples for: ($\Gamma$) rich patients receiving the expensive treatment while poor patients receiving the cheap one – although outcomes of the possible treatments are not particularly dependent on patients' wealth status; ($\Delta$) younger patients receiving surgery while older patients receiving medication; and ($\Upsilon$) genetic information that determines the efficacy of a medication, however, such relationship is unknown to the attending physician.

vational dataset $\mathcal{D}$. The respective graphical model is illustrated in Figure 3. Conforming with the statements above, note that the graphical model also suggests that selection bias is induced by factors $\Gamma$ and $\Delta$, where $\Delta$ represents the confounding factors between $T$ and $Y$.

**Main contribution:** We argue that explicit identification of the underlying factors $\{\,\Gamma, \Delta, \Upsilon\,\}$ in observational datasets offers great insight to guide designing models that better handle selection bias and consequently achieve better performance in terms of estimating ITEs. In this paper, we propose a model, named Disentangled Representations for CounterFactual Regression (DR-CFR), that is optimized to do exactly that. We also present experiments that demonstrate the advantages of this perspective; and show empirically that the proposed method outperforms state-of-the-art models in a variety of data generation scenarios with different dimensionality of factors; see below.

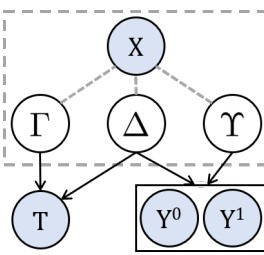

Figure 3: Underlying factors of $X$; $\Gamma$ ($\Upsilon$) are factors that partially determine only $T$ ($Y$) but not the other random variable; and $\Delta$ are confounders; Selection bias is induced by factors $\Gamma$ and $\Delta$.

## 2 RELATED WORKS

Selection bias in observational datasets is equivalent to a domain adaptation scenario where a model is trained on a "source" (observed) data distribution, but should perform well on a "target" (counterfactual) one. Learning treatment effects from observational datasets is closely related to "off-policy learning from logged bandit feedback" – *cf.*, (Swaminathan & Joachims, 2015a), whose goal is learning an optimal policy that selects the best personalized treatment for each individual. A common statistical solution is re-weighting certain data instances to balance the source and target distributions. The majority of re-weighting approaches belong to the Inverse Propensity Weighting (IPW) family of methods – *cf.*, (Austin, 2011; Bottou et al., 2013; Swaminathan & Joachims, 2015c). While IPW methods are unbiased, they suffer from high variance. Swaminathan & Joachims (2015b) proposed the Counterfactual Risk Minimization (CRM) principle to alleviate this issue. In summary, re-weighting is an attempt to **account for** the selection bias.

Johansson et al. (2016) is among the pioneer works that explored ways to use techniques from representation learning (Bengio et al., 2013) to **reduce** the selection bias. Shalit et al. (2017) present a refined version of (Johansson et al., 2016)'s method that learns a common representation space $\Phi(x) = \phi$ by minimizing the discrepancy (Mansour et al., 2009) (hereinafter "disc") between the conditional distributions of $\phi$ given $t = 0$ versus $\phi$, given $t = 1$. That is,

$$\texttt{disc}\Big(\big\{\Phi(x_i)\big\}_{i:\,t_i=0}, \big\{\Phi(x_i)\big\}_{i:\,t_i=1}\Big) \tag{1}$$

which is (effectively) a regularization term that attempts to reduce selection bias in the learned representation. On top of this representation learning network, they trained two regression networks $h^t(\phi)$ – one for each treatment arm ($t \in \{0, 1\}$) – that predict the outcomes.

Hassanpour & Greiner (2019) argued that the learned representation cannot and should not remove all the selection bias, as the confounders not only contribute to choosing a treatment but also to determining the respective outcomes.[2] As a result, where there are confounders (which is a common situation), even $\phi$ would exhibit *some* selection bias, although less than that in the original domain $x$. They built on the work of (Shalit et al., 2017) by introducing *context-aware* importance sampling

---

[2] While Hassanpour & Greiner (2019) presented a graphical model similar to our Figure 3, they only used it to investigate the nature of selection bias. *N.b.*, they did not implement the idea of learning disentangled representations for counterfactual regression; instead, their method [like (Shalit et al., 2017)] learns a common representation $\phi$ that can represent *only* the confounders, but not the other factors. Our approach extends theirs by providing an algorithm that can learn disentangled representations of the underlying factors from observational datasets.

weights, that attempt to account for the above-mentioned remainder selection bias. These weights

$$\omega_i \;=\; 1 + \frac{\Pr(\,\phi_i\,|\,\neg t_i\,)}{\Pr(\,\phi_i\,|\,t_i\,)} \;=\; 1 + \frac{\Pr(\,t_i\,)}{1 - \Pr(\,t_i\,)} \cdot \frac{1 - \pi\big(t_i\,|\,\phi_i\big)}{\pi\big(t_i\,|\,\phi_i\big)} \tag{2}$$

are designed to enhance performance of estimating both factual as well as counterfacual outcomes (by the $1$ and $\frac{\Pr(\,\phi\,|\,\neg t\,)}{\Pr(\,\phi\,|\,t\,)}$ terms, respectively), where $\pi\big(t_i\,|\,\phi_i\big)$ is the probability of assigning the observed $t_i$ conditioned on the learned context $\phi_i$.

Note that both (Shalit et al., 2017) and (Hassanpour & Greiner, 2019) use $\Phi$ to model the concatenation of factors $\Delta$ and $\Upsilon$ (see Figure 3). Although it does make sense that there should be no discrepancy between conditional distributions of $\Upsilon$, the $\Delta$ factor should model the confounding factors, which by definition, must embed some information about treatment assignment. This would result in a positive discrepancy between conditional distributions of $\Delta$ that should not be minimized. Thus, minimizing Equation (1) with respect to $\Phi$ can lead to problematic results as it discards some of the confounders.

Yao et al. (2018) proposed the Similarity preserved Individual Treatment Effect (SITE) method, which extends Shalit et al. (2017)'s framework by adding a local similarity preserving component. This component acts as a regularization term, that attempts to retain the same neighbourhood relationships in the learned representation space as exhibited in the original space, by matching the propensity scores $\Pr(\,t{=}1\,|\,x\,)$ and $\Pr(\,t{=}1\,|\,\phi\,)$. This, however, results in learning sub-optimal representations when $\Gamma \neq \emptyset$ as SITE tries to keep instances whose $\Gamma$s are far apart, also far apart in $\phi$. In other words, this component penalizes reducing selection bias in $\phi$ by not discarding the irrelevant information present in $\Gamma$ even when it does not hurt the outcome estimation at all.

Our work has many similarities to (Kuang et al., 2017), who decomposed $X$ into two subsets: confounding and adjustment variables, which are similar to our $\Delta$ and $\Upsilon$ factors respectively. They then used an optimization algorithm for identifying these variables, to ultimately find an unbiased estimate of the Average Treatment Effect (ATE). We extend their work in three ways: (i) In addition to confounders and adjustment variables, we also identify the factors that determine the treatment and have no effect on the outcome (*i.e.*, $\Gamma$). (ii) Unlike (Kuang et al., 2017) that take a linear approach by tagging the raw features as either confounders or adjustment variables, our proposed method has the capacity to learn [non-linear] representations of the underlying factors. (iii) Our method facilitates estimating both ATE as well ITE, whereas (Kuang et al., 2017) cannot provide estimates of ITEs.

## 3   LEARNING DISENTANGLED REPRESENTATIONS

We assume, without loss of generality, that any dataset of the form $\{\,X,\,T,\,Y\,\}$ is generated from three underlying factors $\{\,\Gamma, \Delta, \Upsilon\,\}$, as illustrated in Figure 3. [3] Observe that the factor $\Gamma$ (resp., $\Upsilon$) partially determines only $T$ (resp., $Y$), but not the other variables; and $\Delta$ includes the confounding factors between $T$ and $Y$. This graphical model suggests that selection bias is induced by factors $\Gamma$ and $\Delta$. It also shows that the outcome depends on the factors $\Delta$ and $\Upsilon$. Inspired by this graphical model, our model architecture incorporates the following components:

- Three representation learning networks; one for each underlying factor: $\Gamma(x)$, $\Delta(x)$, and $\Upsilon(x)$.

- Two regression networks; one for each treatment arm: $h^0(\,\Delta(x), \Upsilon(x)\,)$ and $h^1(\,\Delta(x), \Upsilon(x)\,)$.

- Two logistic networks: $\pi_0\big(t\,|\,\Gamma(x), \Delta(x)\big)$ to model the logging policy – aka behaviour policy in Reinforcement Learning; *cf.*, (Sutton & Barto, 1998) – and $\pi\big(t\,|\,\Delta(x)\big)$ to design weights that account for the confounders' impact.

---

[3] Note that the assumption of unconfoundedness still holds; here is why:
**Short:** Observing either $X$ or $\Delta$ blocks the path from T to Y, which supports the unconfoundedness assumption.
**Long:** Once the representation networks are learned from the observational data, we can compute the latent factors $\{\,\Gamma, \Delta, \Upsilon\,\}$ from $X$ only. Therefore, although these factors are not explicitly observed, they are effectively observed, in that they are derived directly from the observed $X$, and so should not be categorized as "unobserved confounders". For example, the latent factor for "zip code" in $X$ is "socio-economic status" (perhaps in $\Delta$). In other words, "socio-economic status" can be inferred from "zip code" which can be viewed as a proxy for it.

We therefore try to minimize the following objective function:

$$J(\Gamma, \Delta, \Upsilon, h^0, h^1, \pi_0) \;=\; \frac{1}{N}\sum_{i=1}^{N} \omega\big(t_i, \Delta(x_i)\big) \cdot \mathcal{L}\big[\, y_i,\, h^{t_i}\big(\Delta(x_i), \Upsilon(x_i)\big)\,\big] \tag{3}$$

$$+\, \alpha \cdot \texttt{disc}\big(\{\Upsilon(x_i)\}_{i:t_i=0}, \{\Upsilon(x_i)\}_{i:t_i=1}\big) \tag{4}$$

$$+\, \beta \cdot \frac{1}{N}\sum_{i=1}^{N} -\log\big[\, \pi_0\big(t_i \,|\, \Gamma(x_i), \Delta(x_i)\big)\,\big] \tag{5}$$

$$+\, \lambda \cdot \mathfrak{Reg}(\Gamma, \Delta, \Upsilon, h^0, h^1, \pi_0) \tag{6}$$

where $\omega\big(t_i, \Delta(x_i)\big)$ is the re-weighting function; $\mathcal{L}\big[\, y_i,\, h^{t_i}\big(\Delta(x_i), \Upsilon(x_i)\big)\,\big]$ is the prediction loss for observed outcomes (aka factual loss); $\texttt{disc}\big(\{\Upsilon(x)\}_{i:t_i=0}, \{\Upsilon(x)\}_{i:t_i=1}\big)$ calculates the discrepancy between conditional distributions of $\Upsilon$ given $t=0$ versus given $t=1$; $-\log\pi_0(\,\cdot\,)$ is the cross entropy loss of predicting the assigned treatments given the learned context; and $\mathfrak{Reg}(\,\cdot\,)$ is the regularization term for penalizing model complexity. The following sections elaborate on each of these terms.

### 3.1 FACTUAL LOSS: $\mathcal{L}\big[\, y,\, h^t\big(\Delta(x), \Upsilon(x)\big)\,\big]$

Similar to (Johansson et al., 2016; Shalit et al., 2017; Hassanpour & Greiner, 2019; Yao et al., 2018), we train two regression networks $h^0$ and $h^1$, one for each treatment arm. As guided by the graphical model in Figure 3, the inputs to these regression networks are the outputs of the $\Delta(x)$ and $\Upsilon(x)$ representation networks and their outputs are the predicted outcomes for their respective treatments.

Note that the prediction loss $\mathcal{L}$ can only be calculated on the observed outcomes (hence the name *factual loss*), as counterfactual outcomes are not available in any training set. This would be an L2-loss for real-valued outcomes and a log-loss for binary outcomes. By minimizing the factual loss, we ensure that the union of the learned representations $\Delta(x)$ and $\Upsilon(x)$ retain enough information needed for accurate estimation of the observed outcomes.

### 3.2 RE-WEIGHTING FUNCTION: $\omega\big(t, \Delta(x)\big)$

We follow (Hassanpour & Greiner, 2019)'s design for weights as re-stated in Equation (2), with the modification that we employ $\Delta$ to calculate the weights instead of $\Phi$. Although following the same design, we anticipate our weights should perform better in practice than those in (Hassanpour & Greiner, 2019) as: (i) no confounders are discarded due to minimizing the imbalance loss (because our $\texttt{disc}$ is defined based on $\Upsilon$, not $\Phi$); and (ii) only the legitimate confounders are used to derive the weights (*i.e.*, $\Delta$), not the ones that have not contributed to treatment selection (*i.e.*, $\Upsilon$).

Notably, the weights design in Equation (2) is different from the common practice in re-weighting techniques (*e.g.*, IPW) in that the weights are calculated based on all factors that determine $T$ (*i.e.*, $\Gamma$ as well as $\Delta$). However, we argue that incorporation of $\Gamma$ in the weights might result in emphasizing the wrong instances. In other words, since the factual loss $\mathcal{L}$ is only sensitive to factors $\Delta$ and $\Upsilon$, and not $\Gamma$, re-weighting $\mathcal{L}$ according to $\Gamma$ would yield a wrong objective function to be optimized.

### 3.3 IMBALANCE LOSS: $\texttt{disc}\big(\{\Upsilon(x_i)\}_{i:t_i=0}, \{\Upsilon(x_i)\}_{i:t_i=1}\big)$

According to Figure 3, $\Upsilon$ should be independent of $T$ due to the collider structure at $Y$. Therefore,

$$\Upsilon \perp\!\!\!\perp T \quad\Longrightarrow\quad \Pr(\Upsilon\,|\,T) = \Pr(\Upsilon) \quad\Longrightarrow\quad \Pr(\Upsilon\,|\,T{=}0) = \Pr(\Upsilon\,|\,T{=}1) \tag{7}$$

We used Maximum Mean Discrepancy (MMD) (Gretton et al., 2012) to calculate dissimilarity between the two conditional distributions of $\Upsilon$ given $t=0$ versus $t=1$.

By minimizing the imbalance loss, we ensure that the learned factor $\Upsilon$ embeds no information about $T$ and all the confounding factors are retained in $\Delta$. Capturing all the confounders in $\Delta$ and only in $\Delta$ is the hallmark of the proposed method, as we will use it for optimal re-weighting of the factual loss term (next section). Note that this differs from Shalit et al. (2017)'s approach in that they do not distinguish between the independent factors $\Delta$ and $\Upsilon$; and minimizing the loss defined on only one factor $\Phi$ which might erroneously suggest discarding some of the confounders in $\Delta$.

## 3.4 Cross Entropy Loss: $-\log\big[\pi_0\big(t\,|\,\Gamma(x),\Delta(x)\big)\big]$

We model the logging policy as a logistic regression network parameterized by $[\,W_0, b_0\,]$ as follows: $\pi_0(\,t\,|\,\psi\,) = \big[1 + e^{-(\,2t-1\,)(\,\psi\cdot W_0 + b_0\,)}\big]^{-1}$, where $\psi$ is the concatenation of matrices $\Gamma$ and $\Delta$. Minimizing the cross entropy loss enforces learning $\Gamma$ and $\Delta$ in a way that allows $\pi_0(\,\cdot\,)$ to predict the assigned treatments. In other words, the union of the learned representations of $\Gamma$ and $\Delta$ retain enough information to recover the logging policy that guided the treatment assignments.

## 4 Experiments

### 4.1 Benchmarks

Evaluating treatment effect estimation methods is problematic on real-world datasets since, as mentioned earlier, their counterfactual outcomes are inherently unobservable. A common solution is to synthesize datasets where the outcomes of all possible treatments are available, then discard some outcomes to create a proper observational dataset with characteristics (such as selection bias) similar to a real-world one – *cf.*, (Beygelzimer & Langford, 2009; Hassanpour & Greiner, 2018). In this work, we use two such benchmarks: our synthetic series of datasets as well as a publicly available benchmark: the Infant Health and Development Program (IHDP) (Hill, 2011).

#### 4.1.1 Synthetic Datasets

We generated our synthetic datasets according to the following process, which takes as input the sample size $N$; dimensionalities $[m_\Gamma, m_\Delta, m_\Upsilon] \in \mathcal{Z}^{+(3)}$; for each factor $L \in \{\,\Gamma, \Delta, \Upsilon\,\}$, the means and covariance matrices $(\mu_L, \Sigma_L)$; and a scalar $\zeta$ that determines the slope of the logistic curve.

- For each latent factor $L \in \{\,\Gamma, \Delta, \Upsilon\,\}$
  - Form $L$ by drawing $N$ instances (each of size $m_L$) from $\mathcal{N}(\mu_L, \Sigma_L)$,
  - Concatenate $\Gamma$, $\Delta$, and $\Upsilon$ to make the covariates matrix $X$ [of size $N \times (m_\Gamma + m_\Delta + m_\Upsilon)$]
  - Concatenate $\Gamma$ and $\Delta$ to make $\Psi$ [of size $N \times (m_\Gamma + m_\Delta)$]
  - Concatenate $\Delta$ and $\Upsilon$ to make $\Phi$ [of size $N \times (m_\Delta + m_\Upsilon)$]
- For treatment $T$:
  - Sample $m_\Gamma + m_\Delta$ tuple of coefficients $\theta$ from $\mathcal{N}(0, 1)^{m_\Gamma + m_\Delta}$
  - Define the logging policy as $\pi_0(\,t{=}1\,|\,z\,) = \frac{1}{1+\exp(-\zeta z)}$, where $z = \Psi \cdot \theta$
  - For each instance $x_i$, sample treatment $t_i$ from the Bernoulli distribution with parameter $\pi_0(\,t{=}1\,|\,z_i\,)$
- For outcomes $Y^0$ and $Y^1$:
  - Sample $m_\Delta + m_\Upsilon$ tuple of coefficients $\vartheta^0$ and $\vartheta^1$ from $\mathcal{N}(0, 1)^{m_\Delta + m_\Upsilon}$
  - Define $y^0 = (\Phi \circ \Phi \circ \Phi + 0.5) \cdot \vartheta^0 / (m_\Delta + m_\Upsilon) + \varepsilon$ and $y^1 = (\Phi \circ \Phi) \cdot \vartheta^1 / (m_\Delta + m_\Upsilon) + \varepsilon$, where $\varepsilon$ is a white noise sampled from $\mathcal{N}(0, 0.1)$ and $\circ$ is the symbol for element-wise (Hadamard/Schur) product.

We considered all the viable datasets in a mesh generated by $m_\Gamma, m_\Delta, m_\Upsilon \in \{0, 4, 8\}$. This creates 24 scenarios[4] that consider all possible situations in terms of the relative sizes of the factors $\Gamma$, $\Delta$, and $\Upsilon$. For each scenario, we synthesized five datasets with various initial random seeds.

#### 4.1.2 Infant Health and Development Program (IHDP)

The original RCT data was designed to evaluate the effect of specialist home visits on future cognitive test scores of premature infants. Hill (2011) induced selection bias by removing a non-random subset of the treated population to create a realistic observational dataset. The resulting dataset contains 747 instances (608 control, 139 treated) with 25 covariates. We run our experiments on the same benchmark (100 realizations of outcomes) provided by and used in (Johansson et al., 2016; Shalit

---

[4]There are not $2^3 = 27$ scenarios because we removed the three tuples: $(0, 0, 0)$, $(4, 0, 0)$, and $(8, 0, 0)$, as any scenario with $\Delta = \Upsilon = \emptyset$ would generate outcomes that are pure noise.

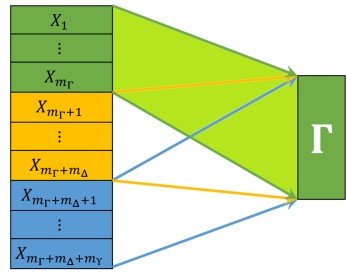
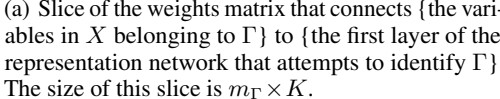
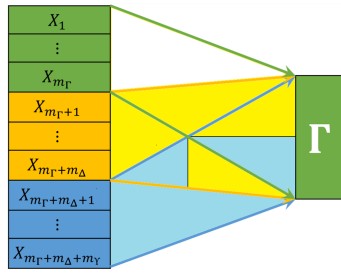

(a) Slice of the weights matrix that connects {the variables in $X$ belonging to $\Gamma$} to {the first layer of the representation network that attempts to identify $\Gamma$}. The size of this slice is $m_\Gamma \times K$.

(b) Slice of the weights matrix that connects {the variables in $X$ *not* belonging to $\Gamma$} to {the first layer of the representation network that attempts to identify $\Gamma$}. The size of this slice is $(m_\Delta + m_\Upsilon) \times K$.

Figure 4: Visualization of slicing the learned weights matrix in the first layer of the representation network (number of neurons: $K$) for identifying $\Gamma$ (best viewed in color).

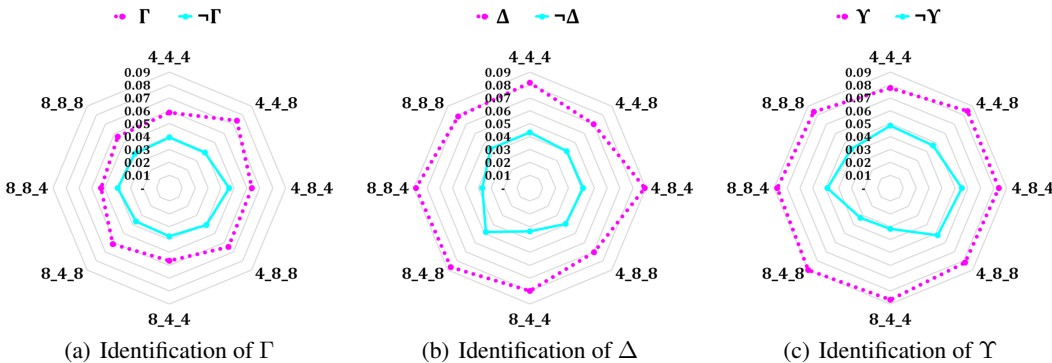

(a) Identification of $\Gamma$      (b) Identification of $\Delta$      (c) Identification of $\Upsilon$

Figure 5: Radar charts that visualize the capability of DR-CFR in identifying the underlying factors $\Gamma$, $\Delta$, and $\Upsilon$. Each vertex on the polygons is identified with the factors' dimension sequence $(m_\Gamma\_m_\Delta\_m_\Upsilon)$ of the associated synthetic dataset. The polygons' radii are scaled between $0:0.09$ and quantify the average weights of the first slice (in dotted magenta) and the second slice (in cyan).

et al., 2017). Outcomes of this semi-synthetic benchmark were simulated according to response surfaces provided in the Non-Parametric Causal Inference (NPCI) package (Dorie, 2016).

## 4.2 RESULTS AND DISCUSSIONS

### 4.2.1 EVALUATING IDENTIFICATION OF FACTORS $\{\Gamma, \Delta, \Upsilon\}$

First, we want to determine if the proposed method is able to identify the variables that belong to each underlying factor. To do so, we look at the weight matrix in the first layer of each representation network, which is of size $(m_\Gamma + m_\Delta + m_\Upsilon) \times K$, where $K$ is the number of neurons in the first hidden layer of the respective representation network. For example, to check if $\Gamma$ is identified properly, we partition the weights matrix into two slices, as shown in Figure 4, and calculate the average of each slice. The first slice [referred to as $S_\Gamma$; highlighted in Figure 4(a)] pertains to "$\Gamma$'s ground truth variables in $X$" and the second slice [$S_{\neg\Gamma}$; Figure 4(b)] pertains to "variables in $X$ that do not belong to $\Gamma$". Constructing $S_\Delta$, $S_{\neg\Delta}$, $S_\Upsilon$, and $S_{\neg\Upsilon}$ follow a similar procedure.

If the proposed method achieves a good identification, then we expect the average of the absolute values of weights in $S_\Gamma$ should be higher than that of $S_{\neg\Gamma}$; this same claim should hold for $(S_\Delta, S_{\neg\Delta})$ and $(S_\Upsilon, S_{\neg\Upsilon})$ as well. Note that only the relative relationships between the average weights in either of the slices matter; since this analysis is aimed at checking whether, for example, for identifying $\Gamma$, its respective representation network has indeed learned to emphasize on "$\Gamma$'s ground truth variables

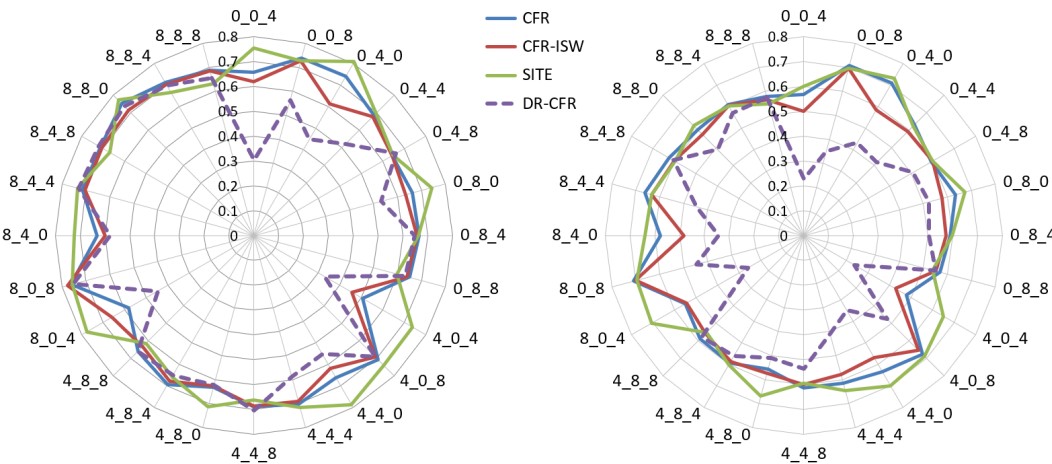

Figure 6: Radar charts for visualizing the PEHE performance results on the synthetic datasets. Training sample size on the left chart is 2,500 and on the right chart is 10,000. Each vertex on the polygons is identified with the factors' dimension sequence ($m_\Gamma\_m_\Delta\_m_\Upsilon$) of the associated group of datasets. The polygons' radii are scaled between $0:0.8$ to quantify the PEHE values (*i.e.*, the closer to the centre, the smaller the PEHE). The dashed purple curve illustrates the results of the proposed method.

in $X$" more than the other variables in $X$. Figure 5 illustrates the identification performance of DR-CFR according to this analysis; showing empirically that the proposed method successfully identifies all the three underlying factors, for all synthetic datasets.

### 4.2.2 EVALUATING ESTIMATION OF TREATMENT EFFECTS

Given a synthetic dataset (that include both factual as well as counterfactual outcomes), one can evaluate treatment effect estimation methods with two types of performance measures:

- Individual-based: "Precision in Estimation of Heterogeneous Effect" $\text{PEHE} = \sqrt{\frac{1}{N}\sum_{i=1}^{N}(\hat{e}_i - e_i)^2}$ where $\hat{e}_i = \hat{y}_i^1 - \hat{y}_i^0$ is the predicted effect and $e_i = y_i^1 - y_i^0$ is the true effect.

- Population-based: "Bias of the Average Treatment Effect" $\epsilon_{\text{ATE}} = \left|\text{ATE} - \widehat{\text{ATE}}\right|$ where $\text{ATE} = \frac{1}{N}\sum_{i=1}^{N}y_i^1 - \frac{1}{N}\sum_{j=1}^{N}y_j^0$ in which $y_i^1$ and $y_j^0$ are the true outcomes for the respective treatments and $\widehat{\text{ATE}}$ is calculated based on the estimated outcomes.

In this paper, we compare performances of the following treatment effect estimation methods: [5]

- **CFR**: CounterFactual Regression (Shalit et al., 2017).
- **CFR-ISW**: CFR with Importance Sampling Weights (Hassanpour & Greiner, 2019).
- **SITE**: Similarity preserved Individual Treatment Effect (Yao et al., 2018).
- **DR-CFR**: Disentangled Representations for CFR – our proposed method.

Figure 6 visualizes the PEHE measures in radar charts for these four methods, trained with datasets of size $N = 2,500$ (left) and $N = 10,000$ (right). As expected, all methods perform better with observing more training data; however, DR-CFR took the most advantage by reducing PEHE the most (by $0.15$, going down from $0.60$ to $0.45$), while CFR, CFR-ISW, and SITE reduced PEHE by $0.07$, $0.08$, and $0.08$ respectively.

Table 1 summarizes the PEHE and $\epsilon_{\text{ATE}}$ measures (lower is better) for all scenarios, in terms of mean and standard deviation of all the $24 \times 5$ datasets, in order to give a unified view on the performance.

---

[5]Note that all four methods share the same core code-base: based on CFR (developed by Johansson et al. (2016) and Shalit et al. (2017)) and so they share very similar model architectures. To allow for fair comparison, we searched their respective hyperparameter spaces, constrained to ensure that all had the same model complexity.

Table 1: Synthetic datasets
($24 \times 5$ with $N = 10{,}000$)

| Methods | PEHE | $\epsilon_{\text{ATE}}$ |
|---|---|---|
| **CFR** | 0.61 (0.05) | 0.021 (0.018) |
| **CFR-ISW** | 0.58 (0.06) | 0.017 (0.009) |
| **SITE** | 0.63 (0.05) | 0.035 (0.039) |
| **DR-CFR** | **0.45 (0.11)** | **0.013 (0.006)** |

Table 2: IHDP datasets
(100 with $N = 747$)

| Methods | PEHE | $\epsilon_{\text{ATE}}$ |
|---|---|---|
| **CFR** | 0.81 (0.30) | 0.13 (0.12) |
| **CFR-ISW** | 0.73 (0.28) | 0.11 (0.10) |
| **SITE** | 0.73 (0.33) | 0.10 (0.09) |
| **DR-CFR** | **0.65 (0.37)** | **0.03 (0.04)** |

PEHE and $\epsilon_{\text{ATE}}$ measures (lower is better) represented in the form of "mean (standard deviation)".

DR-CFR achieves the best performance among the contending methods. These results are statistically significant based on the Welch's unpaired t-test with $\alpha = 0.05$. Table 2 summarizes the PEHE and $\epsilon_{\text{ATE}}$ measures on the IHDP benchmark. The results are reported in terms of mean and standard deviation over the 100 datasets with various realizations of outcomes. Again, DR-CFR achieves the best performance (statistically significant for $\epsilon_{\text{ATE}}$) among the contending methods.

## 5 FUTURE WORKS AND CONCLUSION

The majority of methods proposed to estimate treatment effects – including this work – fall under the category of discriminative approaches. A promising direction is to consider developing generative models, in an attempt to shed light on the true underlying data generating mechanism. Perhaps this could also facilitate generating new, virtual, yet realistic data instances – similar to what is done in computer vision. Louizos et al. (2017)'s method is a notable generative approach, which uses Variational Auto-Encoder (VAE) to extract latent confounders from their observed proxies. While that work is an interesting step in that direction, it is not yet capable of addressing the problem of selection bias. We believe that our proposed perspective on the problem can be helpful to solve this open question. This is left to future work.

In this paper, we studied the problem of estimating treatment effect from observational studies. We argued that not all factors in the observed covariates $X$ might contribute to the procedure of selecting treatment $T$, or more importantly, determining the outcomes $Y$. We modeled this using three underlying sources of $X$, $T$, and $Y$, and showed that explicit identification of these sources offers great insight to help us design models that better handle selection bias in observational datasets. We proposed an algorithm, Disentangled Representations for CounterFactual Regression (DR-CFR), that can (1) identify disentangled representations of the above-mentioned underlying sources and (2) leverage this knowledge to reduce as well as account for the negative impact of selection bias on estimating the treatment effects from observational data. Our empirical results showed that the proposed method achieves state-of-the-art performance in both individual and population based evaluation measures.

## ACKNOWLEDGEMENTS

The authors gratefully acknowledge financial support from Natural Sciences and Engineering Research Council of Canada (NSERC) and Alberta Machine Intelligence Institute (Amii). We wish to thank Dr. Pouria Ramazi and Shivam Raj for fruitful conversations, and Dr. Fredrik Johansson for publishing/maintaining the code-base for the CFR method online. We also would like to thank the ICLR 2020 anonymous reviewers, as well as Dr. Kun Kuang and Tianle Liu, for their valuable reviews, which helped improve this paper.

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
