# OpenReview forum: "Learning Disentangled Representations for CounterFactual Regression"
_ICLR.cc/2020/Conference — Accept (Poster)_

### Official Review · AnonReviewer3 · 2019-10-24
**Official Blind Review #3**

**Rating:** 3

**Review:**


The paper proposes an algorithm that identifies disentangled representation to find out an individual treatment effect. A very specific model that tries to find out the underlying dynamics of such a problem is proposed and is learned by minimizing a suggested objective that takes the strengths of previous approaches. The method is demonstrated in a synthetic dataset and IHDP dataset and shown to outperform other previous methods by a large margin.

My initial review was negative, but I changed my mind after reading a few papers in this area. It seems that explicit learning of underlying factors that are described in (Hassanpour & Greiner, 2019) is a nice idea and works well. My only concern is that the paper has a lot of overlap with (Hassanpour & Greiner, 2019), even using identical figures. I am not sure whether it is OK.



**Experience Assessment:**

I do not know much about this area.

**Review Assessment: Checking Correctness Of Derivations And Theory:**

I assessed the sensibility of the derivations and theory.

**Review Assessment: Checking Correctness Of Experiments:**

I assessed the sensibility of the experiments.

**Review Assessment: Thoroughness In Paper Reading:**

I read the paper at least twice and used my best judgement in assessing the paper.

---

> ### Author Response · Authors · 2019-11-06
> **Thanks! Paper updated with your feedback addressed.**
>
> Thank you for raising your concern. The use of figures similar to those in (Hassanpour & Greiner, 2019) is done under their permission. Please note that these figures are similar but not identical since we generated them.
>
> Regarding the idea of explicit learning of the underlying factors of observational datasets, it is true that (Hassanpour & Greiner, 2019) did use a graphical model that is similar to our Figure 3. However, they only used it to elaborate the problem of selection bias and did not implement the idea. Similar to the work of Shalit et al. (2017), their method learns a common representation $\phi$ that can only be representative of the confounders. Our submission, by contrast, actualizes the idea by providing an algorithm that can in fact learn disentangled representations of the underlying factors, and exploit them to better handle selection bias and consequently achieve better performance in terms of estimating ITEs.
>
> We have updated the paper to address your concerns; see the modified parts (marked as R3, in orange), on page 3. Please forward any other thoughts, ideas, or concerns.

---

### Official Review · AnonReviewer2 · 2019-10-25
**Official Blind Review #2**

**Rating:** 8

**Review:**

The paper proposes a new way of estimating treatment effects from observational data, that decouples (disentangles) the observed covariates X into three sets: covariates that contributed to the selection of the treatment T, covariates that cause the outcome Y and covariates that do both. The authors show that by leveraging this additional structure they can improve upon existing methods in both ITE and ATE

The main contributions of the paper are:
* Highlighting the importance of differentiating between treatment and outcome inducing factors and proposing an algorithm to detect the two
* Creating a joint optimisation model that contains the factual loss, the cross entropy (treatment) loss and the imbalance loss

Overall, I like the paper quite a lot, I find it well-written and clearly motivated with a very nice experimental section that it is designed around understanding the behaviour of the proposed model.

In terms of suggestions, I think it will be very interesting to link the approaches using invariant causal representations with existing work in the Counterfactual Risk Minimization [1] literature and to mutualise the experimental setup.

[1] Swaminathan, Adith, and Thorsten Joachims. "Counterfactual risk minimization: Learning from logged bandit feedback." International Conference on Machine Learning. 2015.

**Experience Assessment:**

I have published in this field for several years.

**Review Assessment: Checking Correctness Of Derivations And Theory:**

I assessed the sensibility of the derivations and theory.

**Review Assessment: Checking Correctness Of Experiments:**

I assessed the sensibility of the experiments.

**Review Assessment: Thoroughness In Paper Reading:**

I read the paper at least twice and used my best judgement in assessing the paper.

---

> ### Author Response · Authors · 2019-11-15
> **Thanks for suggesting an exciting future work!**
>
> Thank you for your thought-provoking suggestion regarding incorporation of the Counterfactual Risk Minimization (CRM) principle [1] into this framework.
>
> In contextual bandits, the goal is often learning an optimal policy that minimizes the regret. There are 2 strategies to do this:
> (i) estimating the outcomes of all possible treatments -- then as a by-product, design the policy such that it selects the treatment promising the best outcome.
> (ii) defining a utility function based on the (importance sampling) weighted observed outcomes to directly learn the policy that optimizes the utility.
> Our goal in this work is estimating the Individual Treatment Effects (ITEs) -- i.e., $Y^1 -Y^0$, not learning an optimal policy $\pi( T | X )$. That is, we want to do (i) but do not care about its by-product.
>
> Looking closer at (Eq. 2), we see that unlike (ii), our method is not stuck with a fixed logger policy $\pi_0( t | x )$ that limits the space for policy search. Instead, we have a tunable denominator $\pi( t | \phi )$ (since $\phi$ is learnable) that can be controlled such that it never becomes too small to de-stabilize the optimization. This however requires coupling the objective function $J( \phi, h )$ (Eqs. 3-6) with the cost function $C( \pi )$ for learning $\pi( t | \phi )$ through $\phi$ (making it $C( \pi, \phi )$ -- FYI, we previously tried this idea but found the optimization did not converge).
>
> A better way would be to employ the CRM principle to enforce learning weights with low variance. We just gave it a quick empirical assessment and the performance seems promising in some cases. This is an exciting future work!
> Thank you very much for this invaluable suggestion and stay tuned for good news! :-)
> We have updated the manuscript to address your insightful comment; see the modified parts marked as R2 on page 3, in cyan.

---

### Official Review · AnonReviewer1 · 2019-11-07
**Official Blind Review #1**

**Rating:** 8

**Review:**

Summary:
   The authors consider the problem of estimating average treatment effects when observed X and treatment T causes Y. Observational data for X,T,Y is available and strong ignorability is assumed. Previous work (Shalit et al 2017) introduced learning a representation that is invariant in distribution across treatment and control groups and using that with treatment to estimate Y. However, authors point out that this representation being forced to be invariant still does not drive the selection bias to zero. A follow up work (Hassanpour and Greiner 2019) - corrects for this by using additional importance weighting that estimates the treatment selection bias given the learnt representation. However, the authors point out even this is not complete in general, as X could be determined by three latent factors, one that is the actual confounder between treatment and outcome and the other that affects only the outcome and the other that affects only the treatment. Therefore, the authors propose to have three representations and enforce independence between representation that solely determines outcome and the treatment and make other appropriate terms depend on the respective latent factors. This gives a modified objective with respect to these two prior works.

The authors implement optimize this joint system on synthetic and real world datasets. They show that they outperform all these previous works because of explicitly accounting for confounder, latent factors that solely control only outcome and treatment assignment respectively.

Pros:
  This paper directly addresses the problems due to Shalit 2017 that are still left open. The experimental results seems convincing on standard benchmarks.

I vote for accepting the paper. I don't have many concerns about this paper.

Cons:
  - I have one question for the authors - if T and Y(0),Y(1) are independent given X is assumed, then how are we sure that the composite representations (of the three latent factors) are going to necessarily satisfy ignorability provably ?? I guess this cannot be formally established. It would be great for the authors to comment on this.




**Experience Assessment:**

I have read many papers in this area.

**Review Assessment: Checking Correctness Of Derivations And Theory:**

N/A

**Review Assessment: Checking Correctness Of Experiments:**

I assessed the sensibility of the experiments.

**Review Assessment: Thoroughness In Paper Reading:**

I read the paper at least twice and used my best judgement in assessing the paper.

---

> ### Author Response · Authors · 2019-11-14
> **Thanks! Paper updated with your feedback addressed.**
>
> Thank you for asking this important, thought-provoking question -- does  the ignorability assumption still hold in the presence of the three latent factors?  More precisely, is $T \perp Y  |  X$ -- i.e., are there no active paths in the Belief Net (Figure 3) connecting $T$ to $Y$, given $X$?
>
> The answer is Yes:
> Once the model (here, representation networks) is learned from the training data $\{ [ x_i, y_i, t_i ] \}_{i=1..N}$, we can then compute the latent factors $\{ \Gamma, \Delta, \Upsilon \}$ from $X$ only. Therefore, although these factors are not explicitly observed, they are effectively observed, in that they are derived directly from the observed $X$, and so should not be categorized as “unobserved confounders”. For example, the latent factor for “zip code” in $X$ is “socio-economic status” (perhaps in $\Delta$). In other words, “socio-economic status” can be inferred from “zip code” which can be viewed as a proxy for it. Observing either $X$ or $\Delta$ blocks the path from $T$ to $Y$, which supports the assumption of unconfoundedness.
>
> Unconfoundedness (and consequently ignorability) only breaks when we have truly unobserved confounders (let’s refer to them as $U$ -- in contrast with the essentially-observed confounders that are derived from $X$). For example, the doctor looks at the patient’s MRI scan ($U$) and based on that prescribes surgery ($T$); note that the information present in the scan will also affect the patient's outcome ($Y$). However, as the scan is not filed in the patient’s health records, it is not part of $X$, and so it is not observed.
>
> We have updated the manuscript to address your valuable comment; see the modified parts in the paper, marked as R1, in green:
> (i) Figure 3 on page 3
> (ii) Footnote on page 4

---

### Public Comment · ~Kun_Kuang1 · 2019-10-28
**Comparing with a similar paper [1]**

The idea of this paper is similar to a previous AAAI paper [1], which proposed to decompose all observed variables into different subsets, including confounders, adjustment variables. The decomposed confounders are for removing selection bias, while the adjustment variables are used for regression adjustment. But [1] was proposed for estimating the average treatment effect.

I think it would be better if the authors can acknowledge the previous paper [1],  introduce the difference between their paper and [1] in a few words, and reclaim their contributions.

[1]. Kuang K, Cui P, Li B, et al. Treatment effect estimation with data-driven variable decomposition[C]//Thirty-First AAAI Conference on Artificial Intelligence. 2017.

---

> ### Author Response · Authors · 2019-11-04
> **Thanks; will do.**
>
> We would like to thank Dr. Kuang for bringing this interesting work to our attention. Similar to our work, [1] also proposed to decompose observed variables essentially into two subsets: confounding (referred to as $X$) and adjustment ($Z$) variables that are equivalent to our $\Delta$ and $\Upsilon$ factors. [1] then proposed an optimization algorithm to identify $X$ and $Z$ which were later used to find an unbiased estimate of the Average Treatment Effect (ATE).
>
> However, there are several important differences between [1] and our proposed method:
> 1. In addition to confounders and adjustment variables, we also identify the factors that determine the treatment and have no effect on the outcome; we refer to them as $\Gamma$. Although such factors do induce selection bias, they should not be considered in re-weighting samples when trying to estimate counterfactual outcomes. Therefore, identifying these terms, and then explicitly removing their influence is important.
> 2. Another major difference is that our proposed method facilitates estimating both ATE as well as Individual Treatment Effect (ITE). It is unclear, however, if [1] could provide estimates of ITEs since $h(U)$ does not accept treatment as its input.
>
> We will certainly mention and cite [1] in our updated version of the paper and include a brief discussion on the similarities and differences.

---

> > ### Author Response · Authors · 2019-11-06
> > **Thanks! Paper updated with your feedback addressed.**
> >
> > We have updated the manuscript to address your valuable comment.; see the modified material marked as RK, in pink, page 4.

---

### Public Comment · ~Tianle_Liu1 · 2019-12-25
**How to distinguish $\Upsilon$ from $\Delta$ in your algorithm?**

Thanks for the nice paper and merry Christmas! Really enjoyed it. However, I have a question here:

It seems if you replace $\Delta$ by $(\Delta, \Upsilon)$ and $\Upsilon$ by $0$, the loss would always reduce since the discrepancy term reduces and the other terms do not change. So how could you distinguish $\Upsilon$ from $\Delta$ by minimizing the objective function?

---

> ### Author Response · Authors · 2020-01-14
> **A bottle-neck implicitly derives the optimization procedure to learn to distinguish $\Delta$ from $\Upsilon$.**
>
> Thank you for your thought-provoking comment.
>
> First of all, the regularization term
> $Reg( \Gamma, \Delta, \Upsilon, h^0, h^1, \pi_0 )$
> assures that the intersection between each pair of the representations learned for $\Gamma$, $\Delta$, and $\Upsilon$ is empty.
>
> The discrepancy term ensures that the representation learned for $\Upsilon$ ( let us call it $Rep(\Upsilon)$ ) embeds no information about the confounders $\Delta$; and only embeds information about $\Upsilon$ factors, if any. However, based on the objective function in Equations (3-6), it appears that there are no explicit constraints on what information $Rep(\Delta)$ can learn to embed: it could be $\Delta$ and even some $\Upsilon$.
>
> Your question is about the extreme case: why optimization does not lead to learning a $Rep(\Upsilon)$ that embeds no information (i.e., learns only noise) and a $Rep(\Delta)$ that learns to embed both $\Delta$ and $\Upsilon$.
>
> We don’t have a theoretical proof that this won’t occur, but have a hypothesis that is also supported by empirical evidence. As mentioned earlier, the discrepancy term works as a sieve, allowing only the $\Upsilon$ factors (ones that are related to only Y, but not X nor T) to be learned by $Rep(\Upsilon)$. If all $\Upsilon$ are represented by $Rep(\Upsilon)$, then due to regularization, $Rep(\Delta)$ will not represent any $\Upsilon$ and would only represent $\Delta$. This is the desired case. However, if some $\Upsilon$ are left out and not represented by $Rep(\Upsilon)$, $Rep(\Delta)$ must compensate and represent them, which effectively will reduce the capacity of $Rep(\Delta)$ to represent $\Delta$ -- i.e., factors that cannot be represented by any of the other components.
>
> Our hypothesis is that this bottle-neck implicitly derives the optimization procedure to learn to distinguish $\Delta$ from $\Upsilon$.
>
> We empirically tested this hypothesis by limiting the capacity of the $Rep(\Upsilon)$ network (by reducing the number of hidden neurons -- from 30 to 15 to 0) while increasing the capacity of $Rep(\Delta)$ (from 30 to 45 to 60 respectively). Our performance results are best at the (30, 30) setting and worsen as the capacity of $Rep(\Upsilon)$ is decreased, regardless of whether the capacity of $Rep(\Delta)$ is increased. A particularly important observation is that the increase in the capacity of $Rep(\Delta)$ does not replace the need for having a separate $Rep(\Upsilon)$ network: It must be $Rep(\Upsilon)$ that embeds $\Upsilon$, not $Rep(\Delta)$.
>
> See the Figures here: http://webdocs.cs.ualberta.ca/~hassanpo/misc/rev_Liu.PNG

---

### Decision · Program_Chairs · 2019-12-19

**Decision:**

Accept (Poster)

**Comment:**

The paper proposes a new way of estimating treatment effects from observational data. The text is clear and experiments support the proposed model.